# Methionine Cycle Rewiring by Targeting miR-873-5p Modulates Ammonia Metabolism to Protect the Liver from Acetaminophen

**DOI:** 10.3390/antiox11050897

**Published:** 2022-04-30

**Authors:** Rubén Rodríguez-Agudo, Naroa Goikoetxea-Usandizaga, Marina Serrano-Maciá, Pablo Fernández-Tussy, David Fernández-Ramos, Sofía Lachiondo-Ortega, Irene González-Recio, Clàudia Gil-Pitarch, María Mercado-Gómez, Laura Morán, Maider Bizkarguenaga, Fernando Lopitz-Otsoa, Petar Petrov, Miren Bravo, Sebastiaan Martijn Van Liempd, Juan Manuel Falcon-Perez, Amaia Zabala-Letona, Arkaitz Carracedo, Jose Vicente Castell, Ramiro Jover, Luis Alfonso Martínez-Cruz, Teresa Cardoso Delgado, Francisco Javier Cubero, María Isabel Lucena, Raúl Jesús Andrade, Jon Mabe, Jorge Simón, María Luz Martínez-Chantar

**Affiliations:** 1Liver Disease Laboratory, Center for Cooperative Research in Biosciences (CIC bioGUNE), Basque Research and Technology Alliance (BRTA), Bizkaia Technology Park, Building 801A, 48160 Derio, Spain; rrodriguez@cicbiogune.es (R.R.-A.); ngoikoetxea@cicbiogune.es (N.G.-U.); mserrano@cicbiogune.es (M.S.-M.); pablo.fernandeztussy@yale.edu (P.F.-T.); dfernandez.ciberehd@cicbiogune.es (D.F.-R.); slachiondo@cicbiogune.es (S.L.-O.); irecio@cicbiogune.es (I.G.-R.); cgil@cicbiogune.es (C.G.-P.); mmercado@cicbiogune.es (M.M.-G.); mbizcarguenaga@cicbiogune.es (M.B.); flopitz@cicbiogune.es (F.L.-O.); petar.petrov@ciberehd.es (P.P.); mbravo@cicbiogune.es (M.B.); amartinez@cicbiogune.es (L.A.M.-C.); tcardoso@cicbiogune.es (T.C.D.); 2Centro de Investigación Biomédica en Red de Enfermedades Hepáticas y Digestivas (CIBERehd), 28029 Madrid, Spain; jose.castell@uv.es (J.V.C.); ramiro.jover@uv.es (R.J.); fcubero@ucm.es (F.J.C.); lucena@uma.es (M.I.L.); andrade@uma.es (R.J.A.); 3Precision Medicine and Metabolism Laboratory, Center for Cooperative Research in Biosciences (CIC bioGUNE), Basque Research and Technology Alliance (BRTA), Bizkaia Technology Park, Building 801A, 48160 Derio, Spain; 4Department of Immunology, Ophthalmology and ENT, Complutense University School of Medicine, Instituto de Investigación Sanitaria Gregorio Marañon (IiSGM), 28040 Madrid, Spain; lmoran@ucm.es; 5Unidad de Hepatología Experimental, Health Research Institute Hospital La Fe, Av. Fernando Abril Martorell, 46026 Valencia, Spain; 6Departamento de Bioquímica y Biología Molecular, Facultad de Medicina, Universidad de Valencia, Av. de Blasco Ibáñez 15, 46010 Valencia, Spain; 7Metabolomics Platform, Center for Cooperative Research in Biosciences (CIC bioGUNE), Basque Research and Technology Alliance (BRTA), Bizkaia Technology Park, Building 801A, 48160 Derio, Spain; smvanliempd@cicbiogune.es (S.M.V.L.); jfalcon@cicbiogune.es (J.M.F.-P.); 8Ikerbasque, Basque Foundation for Science, 48013 Bilbao, Spain; acarracedo@cicbiogune.es; 9Cancer Cell Signaling and Metabolism Laboratory, Center for Cooperative Research in Biosciences (CIC bioGUNE), Basque Research and Technology Alliance (BRTA), Bizkaia Technology Park, Building 801A, 48160 Derio, Spain; azabala@cicbiogune.es; 10Centro de Investigación Biomédica en Red de Cáncer (CIBERONC), Instituto Carlos III, 28029 Madrid, Spain; 11Traslational prostate cancer Research Lab, CIC bioGUNE-Basurto, Biocruces Bizkaia Research Health Institute, 48903 Barakaldo, Spain; 12Servicio de Farmacología Clínica, Instituto de Investigación Biomédica de Málaga—IBIMA, Hospital Universitario Virgen de la Victoria, Universidad de Málaga, 29010 Malaga, Spain; 13UICEC IBIMA, Plataforma ISCiii de Investigación Clínica, 28020 Madrid, Spain; 14Unidad de Gestión Clínica de Enfermedades Digestivas, Instituto de Investigación Biomédica de Málaga-IBIMA, Hospital Universitario Virgen de la Victoria, Universidad de Málaga, 29010 Malaga, Spain; 15IK4-Tekniker, 20600 Eibar, Spain; jon.mabe@tekniker.es

**Keywords:** drug-induced liver injury (DILI), acetaminophen (APAP), ammonia, methionine cycle, miR-873-5p, therapy, polyamines, mitochondria

## Abstract

Drug-induced liver injury (DILI) development is commonly associated with acetaminophen (APAP) overdose, where glutathione scavenging leads to mitochondrial dysfunction and hepatocyte death. DILI is a severe disorder without effective late-stage treatment, since N-acetyl cysteine must be administered 8 h after overdose to be efficient. Ammonia homeostasis is altered during liver diseases and, during DILI, it is accompanied by decreased glycine N-methyltransferase (GNMT) expression and S-adenosylmethionine (AdoMet) levels that suggest a reduced methionine cycle. Anti-miR-873-5p treatment prevents cell death in primary hepatocytes and the appearance of necrotic areas in liver from APAP-administered mice. In our study, we demonstrate a GNMT and methionine cycle activity restoration by the anti-miR-873-5p that reduces mitochondrial dysfunction and oxidative stress. The lack of hyperammoniemia caused by the therapy results in a decreased urea cycle, enhancing the synthesis of polyamines from ornithine and AdoMet and thus impacting the observed recovery of mitochondria and hepatocyte proliferation for regeneration. In summary, anti-miR-873-5p appears to be an effective therapy against APAP-induced liver injury, where the restoration of GNMT and the methionine cycle may prevent mitochondrial dysfunction while activating hepatocyte proliferative response.

## 1. Introduction

The liver is the main organ responsible for maintaining an adequate nitrogen balance in the organism. The combination of ammonia-scavenging pathways guarantees a well-balanced pH regulation and homeostasis of the mitochondrial urea cycle in the periportal region and glutamine synthetase (GLUL) in the perivenous region [1]. Nevertheless, hyperammonemia is a metabolic condition widely linked to acute and chronic liver diseases [2,3], in which a dysregulation in either the urea cycle and/or GLUL leads to perturbations in the homeostasis of the cation. Related to this, our group previously developed a method for staining hepatic ammonia that allows for characterizing hyperammonemia in chronic liver diseases, such as non-alcoholic fatty liver disease (NAFLD) or cirrhosis, and in acute liver injury caused by acetaminophen (APAP) overdose [4].

Acute liver failure is a severe disorder mainly present in Western countries and caused by the hepatotoxic effect of drugs [5], where drug-induced liver injury (DILI) is the term normally used to define the condition. With the majority of cases being related to APAP, in the USA, the overdosing of this drug is the leading cause of liver failure [6], affecting 14 per 100,000 people [7] and leading annually to 500 deaths, 50,000 visits to the emergency room and 10,000 hospitalizations [8,9]. Furthermore, in Europe, APAP overdose comprises 40−70% of all DILI cases [10]. Currently, the most common treatment for APAP-related liver failure consists of N-acetyl cysteine (NAC) administration, with a probability of 66% for rescuing the liver only if early treatment is provided 8 h after intoxication [11,12]. Overall, current therapeutic approaches offer low chances of rescuing the liver from DILI, including the mainstays of early diagnosis and the removal of the suspected drug [13,14]. Thus, new approaches are required to improve the prognosis of these patients.

Since their recent approval by the Food and Drugs Agency (FDA), small RNA molecules have entered clinical practice. In this context, a growing number of reports have suggested the significant utility of micro-RNAs (miRNAs) as either biomarkers or drugs, enhancing their use in medical intervention for many diseases [15]. In this context, a large number of molecules, including miR-34a/b, miR-200c and miR-378, has been related to liver disorders, including fibrosis and viral hepatitis [16]. Several therapies with compounds targeting miR-34 and miR-122 are currently being evaluated in clinical trials for liver cancer or hepatitis C virus treatment [15]. Interestingly, our group previously highlighted the potential contribution of miR-873-5p to liver diseases, characterizing its upregulation in NAFLD [17] and fibrosis [18]. In these works, miR-873-5p was reported to be an epigenetic regulator of glycine N-methyltransferase (GNMT) for inhibiting its expression.

GNMT is a cytosolic, nuclear and mitochondrial enzyme within hepatocytes that participates in the methionine cycle catalyzing the conversion of S-adenosyl-methionine (AdoMet) to S-adenosyl-homocysteine, being the main enzyme in the liver responsible for the catabolism of excess hepatic AdoMet and synthesis of sarcosine [19]. The activity of the methionine cycle is essential for correct liver functioning, whereas perturbations in this pathway have been widely linked to liver pathologies [19,20]. The intricate metabolic network in the hepatocyte connects the methionine cycle to other pathways such as the folates cycle [21], the urea cycle [22] or polyamine synthesis [23], among others. The modulation of a certain pathway redirects the metabolic flux towards others [24]. Taking this into consideration, the GNMT expression and methionine cycle recovery mediated by miR-873-5p targeting may modulate the metabolic efflux in the hepatocyte and have an impact on APAP overdose pathophysiology.

The present work evaluates perturbations in ammonia metabolism in the APAP derived hepatotoxicity during DILI. Basing on the existing connection between ammonia homeostasis and the methionine cycle, we demonstrate that GNMT recovery under miR-873-5p knockdown induces a shift from the urea cycle towards polyamine synthesis, preventing DILI development in hepatocytes.

## 2. Materials and Methods

### 2.1. Animal Maintenance and Preclinical Studies

All procedures were approved by the CIC bioGUNE Animal Care and Use Committee and the local authority (Diputación de Bizkaia), under the codes P-CBG-CBBA-0218 and P-CBG-CBBA-1421, according to the criteria established by the European Union. Three-month-old C57BL/6J male mice were maintained with ad libitum access to water and a standard chow diet. They were administered 360 mg/kg APAP dissolved in PBS through intraperitoneal injection. Mice were treated 24 h after injection (see below) and sacrificed 24 h later at a final 48 h endpoint. Samples of liver for cryopreservation and paraffin- or O.C.T-embedded and serum were collected.

### 2.2. Treatment of Primary Mouse Hepatocytes

Primary mouse hepatocytes were obtained from 3-month old C57BL/6J or *Glycine-N-methyltransferase* lacking (*Gnmt*^−/−^) mice by perfusion with Type I collagenase. Hepatocytes were washed three times with Minimal Essential Medium (MEM, ThermoFisher Scientific, Waltham, MA, USA) containing 10% FBS (ThermoFisher Scientific), 1% PSA-G (penicillin-streptamycin-antimycin and glutamine, ThermoFisher Scientific). Cells were seeded over previously collagen I (Corning Inc., Corning, NY, USA)-coated culture plates in MEM 10% FBS, 1% PSA-G. Upon attachment, isolated mouse primary hepatocytes were transfected by overnight incubation with 25 nM anti-miR-873-5p/miR-Ctrl/mimic-miR-873-5p (Horizon Discovery, Waterbeach, Cabridgeshire, UK) using DharmaFECT (GE Healthcare Dharmacon Inc., Lafayette, CO, USA) 1 or jetPRIME (Polyplus-transfection, Strasbourg, Illkrich-Graffenstaden, France). After removing transfection medium 6 h after transfection, hepatocytes were maintained overnight in MEM 0% FBS 1% PSA-G and treated next day with 10mM APAP. Finally, hepatocytes were collected at different times: 0, 1, 3 and 6 h. Different treatments were administered to primary hepatocytes 30 min prior APAP administration: 0.5 μM Dimethylfluoroornithine (DFMO, Sigma-Aldrich, St. Louis, MO, USA), 1 μM SAM486A (Sigma-Aldrich) and 1−2.5 mM ammonium chloride (Sigma-Aldrich) were used.

### 2.3. Human Subjects

Measurements of serum miR-873-5p and AST, ALT, ALP and TBL were performed in different cohorts recruited at the Hospital Universitario Virgen de la Victoria, Málaga, Spain (Table 1). Ten human serum samples (six males and four females) with idiosyncratic Drug Induced Liver Injury (DILI) and ten controls (five males and five females) were evaluated for micro-RNA expression levels. Serum samples were obtained at the time of DILI recognition. All the procedures were approved by the Research Ethics Committee of Malaga Hospital (Code AND-HEP-2015-01). The investigators endorse that all patients gave informed consent for the clinical studies, according to the principles embodied in the Declaration of Helsinki.

### 2.4. miR-873-5p Targeting In Vivo

Mice treated with 360 mg/kg APAP (Sigma-Aldrich) were divided into two groups (*n* = 4) and administered 60 μg/mouse of an anti-miR-873-5p or miR-Ctrl using Invivofectamine^®^ 3.0 (Invitrogen) Reagent through tail vein injection, which allows a specific silencing in the liver. Mice were sacrificed after 48 h of APAP administration. Samples of serum and liver for cryopreservation and paraffin- or O.C.T-embedded were collected.

### 2.5. Histological Procedures

All the samples were cut with HistoCore MULTICUT microtome, then deparaffined with Histo-Clear I solution (Fisher Scientific, Hampton, NH, USA) and hydrated through a decreasing concentration of alcohol solutions. Several stainings have been performed: hematoxylin-eosin (H&E) (Sigma-Aldrich), ammonia and immunohistochemistry for glutamine synthetase (GLUL), F4/80, cyclin D1 and proliferating cell nuclear antigen (PCNA). H&E staining: 5 μm sections were subjected to conventional hematoxylin and eosin staining, dehydrating samples and clearing them with histoclear before DPX permanent mounting. Ammonia: 5 μm samples of the paraffin tissue array were incubated for 5 min with 100 mL of Nessler’s reagent (Fisher Scientific) and washed for 10 s with sterile distilled water. Samples were counterstained with Mayer’s hematoxylin, washed with water and dehydrated briefly before clearing with histoclear. Samples were mounted with DPX permanent mounting medium. Nessler’s reagent becomes darker yellow in the presence of ammonia, forming precipitates at higher concentrations. IHC: 5 μm sections were unmasked with 15′ Proteinase K at RT according to the primary antibody used (Appendix A) and subjected to endogenous peroxide blocking (3% H_2_O_2_ in PBS, 10′, RT). For mouse-hosted antibodies, samples were blocked with 2.5% goat anti-mouse Fab fragment (1:10, 1 h, RT) and then blocked with 5% goat serum (30′, RT). Sections were incubated in a humid chamber with the primary antibody in antibody diluent (2% BSA with 0.01% PBS-azide, 1 h, RT) followed by ImmPRESS HRP-conjugated secondary antibodies for Rabbit (Cyclin D1 and GLUL), Rat (F4/80) and Mouse (PCNA) (Vector Laboratories Inc., Burlingame, CA, USA) for 30′ and RT. Colorimetric detection was confirmed with Vector VIP chromogen (Vector Laboratories Inc.) and sections were counterstained with hematoxylin prior to mounting with DPX mounting medium. Images were captured using Leica DM750 optical microscopy with a digital color camera (Leica ICC50W), obtaining 5–10 random images per sample. Stained area percentages of each sample were calculated using FIJI (ImageJ) https://imagej.net/Fiji, accessed on 10 March 2022.

### 2.6. RNA Isolation and Quantitative Real-Time Polymerase Chain Reaction

Total RNA from liver and primary hepatocytes was isolated with Trizol (Invitrogen, Waltham, MA, USA). One to two µg of total RNA were treated with DNAse (Invitrogen) and reverse transcribed into cDNA using M-MLV Reverse Transcriptase (Invitrogen). Quantitative real-time PCR (qPCR) was performed using SYBR^®^ Select Master Mix (Applied Biosystems, Waltham, MA, USA) and the Viia7/QS6 Real-Time PCR System (Applied Biosystems). The Ct values were compared with a certain group (Ctrl or APAP + siCtrl), and data were then normalized to the housekeeping expression of *Arp*. Primers sequences are described in Appendix A.

### 2.7. Metabolite Analysis

Metabolites from tissue/serum were extracted in methanol/water (50/50% *v*/*v*) with 10 mM acetic acid and 10 μM stable labelled 13CD3-methionine (methionine-SL) (Cambridge Isotope Laboratories, Tewksbury, MA, USA) as internal standard. Chilled extracts were evaporated with a SpeedVac for approximately 2 h. Pellets were dissolved in water/acetonitrile (MeCN)/formic acid (FA) (39.9/60/0.1% *v*/*v*/*v*). Samples were measured with a UPLC system (Acquity, Waters Inc., Manchester, UK) coupled to a Time of Flight mass spectrometer (ToF MS, SYNAPT G2, Waters Inc.). In a 2.1 × 100 mm, 1.7 μm BEH amide column (Waters Inc.), samples were separated in different solvents (previously stabilized at 40 °C): solvent A (aqueous phase) consisted of 99.5% water, 0.5% FA and 20 mM ammonium formate, while solvent B (organic phase) consisted of 29.5% water, 70% MeCN, 0.5% FA and 1 mM ammonium formate. Extracted ion traces were obtained for AdoMet (*m/z* = 3 99.145), 13CD3-methionine (*m/z* = 154.0796), spermine (*m/z* = 203.2236), spermidine (*m/z* = 146.1657), GSH (*m/z* = 308.0916) and GSSG (*m/z* = 613.1598) in a 20 mDa window for the most abundant isotopes and subsequently smoothed and integrated with QuanLynx software (Waters, Manchester, UK). Metabolite levels were normalized using mg of liver tissue taken.

### 2.8. Protein Isolation and Western Blotting

Total protein extracts from primary hepatocytes and hepatic tissue were resolved in sodium dodecyl sulfate-polyacrylamide gels and transferred to nitrocellulose membranes (GE Healthcare). After 1 h blocking with 5% milk diluted in tris-buffer saline with 0.1% tween-20 (TBS-T) at room temperature, membranes were incubated overnight at 4 °C with a primary antibody diluted in 5% milk in TBS-T. As secondary antibodies, we used anti-rabbit-IgG-HRP-linked (Cell Signaling Technology, Danvers, MA, USA) and anti-mouse IgG-HRP-linked (Cell Signaling) diluted in 5% milk in TBS-T for 1 h at room temperature. The antibodies and conditions used for Western Blotting are described in Appendix A.

### 2.9. Hepatic microRNA Quantitative Real-Time PCR

Total RNA was from liver and primary hepatocytes isolated with Trizol (Invitrogen) using TaqMan^®^ MicroRNA Reverse Transcription Kit (Thermo Fisher Scientific, 4366597) procedure using 100 ng of total RNA and using specific oligos for hsa-miR-873-5p (Thermo Fisher Scientific, 002356) and U6 (Thermo Fisher Scientific, 001973). Quantitative real-time PCR was performed using TaqMan^®^ Universal PCR Master Mix, no AmpErase^®^ (Thermo Fisher Scientific, 432408), following manufacturer’s procedure. Expression levels were normalized using U6 snRNA. Sequences are described in Appendix A.

### 2.10. Serum microRNA Quantification

Micro-RNAs were isolated from serum with the miRNeasy Serum/Plasma Kit (QIAGEN, Hilden, Mettmann, Germany) following manufacturer’s procedure, adding miR-39 as a Spike-In Control. miR-873-5p-specific RT/q-PCR were performed as described and miR-873-5p levels were normalized to the Spike-In.

### 2.11. TUNEL Assay for Cell Death Detection

Apoptotic cells were detected by TUNEL staining using in situ cell death detection kit (Roche, city, State, country). Cells were fixed in 4% paraformaldehyde for 10 min, washed twice in PBS and treated with 3% H_2_O_2_ (Panreac Applichem, Darmstadt, Germany) diluted in MeOH (Merk Millipore, Basilea, Schwitzerland) for 5 min and citrate pH 7.4 for 3 min, then, cells were treated with an enzyme:FITC:buffer mixture (1:9:40) and mounted with Fluoroshield^TM^ with DAPI (Sigma-Aldrich). Five random images per sample were taken and the percentage of TUNEL positive cells was calculated using FIJI (FIJI is just ImageJ) https://imagej.net/Fiji, accessed on 10 March 2022.

### 2.12. GNMT Immunofluorescence

Cells previously fixed in 4% paraformaldehyde (ThermoFisher Scientific, J19943) were blocked and permeabilized with 0.1% BSA (Sigma-Aldrich, A3912), 10% GS (Sigma) with 0.01% Triton X-100 (Sigma-Aldrich, T9284). Then, samples were incubated overnight at 4 °C with anti-GNMT homemade antibody diluted 1:1000 in 0.1% BSA and 10% GS. The next day, samples previously washed with PBS were developed with secondary Rabbit-Cy3 (1:200; 1 h; RT in 0.1% BSA and 10% GS). Pictures were taken with an Axio Imager D1 Upright Fluorescence Microscope (Carl Zeiss AG, Jena, Germany).

### 2.13. Transaminases Determination in Serum

The levels of aspartate aminotransferase (AST) and alanine aminotransferase (ALT) in serum were determined by a Selectra Junior Spinlab 100 analyzer (Vital Scientific, Dieren, The Netherlands) according to the manufacturers’ suggested protocol.

### 2.14. Enzyme-Linked Immunosorbent Assay (ELISA)

The amount of serum tumor necrosis factor (TNF) in serum was measured using 20 µL of samples by paired antibodies. Samples were analyzed by ELISA using the DuoSet II kit (R&D Systems, Minneapolis, MN, USA) according to the manufacturer’s recommendations.

### 2.15. SOD Assay in Liver Homogenates

Superoxide dismutase activity was followed through a colorimetric assay by using a SOD Determination Kit (Sigma-Aldrich). The procedure was realized according to the manufacturer’s instructions.

### 2.16. Determination of Mitochondrial Reactive Oxygen Species (ROS)

Mitochondrial ROS production in primary hepatocytes was assessed using MitoSOX Red mitochondrial superoxide indicator (ThermoFisher Scientific). The cells were loaded with 2 μM MitoSOX Red for 10 min at 37 °C in a CO_2_ incubator. The cells were then washed three times with PBS. Fluorescence was read at 510 nm (excitation) and 595 nm (emission) using a plate reader SpectraMax M2 (bioNova Científica SL, Madrid, Comunidad de Madrid, Spain).

### 2.17. Respiration Studies in Primary Hepatocytes

The respiration activity in primary hepatocytes was measured at 37 °C by high-resolution respirometry with the Seahorse Bioscience XF24-3 Extracellular Flux Analyzer. Succinate (10 mM) and rotenone (2 μM) were used as substrates to quantify State 2. State 3 was initiated with ADP, State 4 induced with the addition of oligomycin (State 4o), and FCCP-induced maximal uncoupler-stimulated respiration (State 3u) were sequentially measured. The normalized data are expressed as pmol of O_2_ per minute or milli-pH units (mpH) per minute, per µg total protein. The dual-ATP production rate was assessed using Seahorse XFe24 Analyzer (Agilent Technologies, Santa Clara, CA, USA); simultaneous reads of ATP production from glycolysis and mitochondria were performed using a label-free technology XF Real-Time ATP Rate Assay kit, as described in the User Guide (Agilent Technologies).

### 2.18. Mitochondrial Labelling

The relative number of functional mitochondria was determined by using a MitoTracker^TM^ Green FM (ThermoFisher) probe. Cells were stained according to manufacturer’s instructions and then fixed with 4% paraformaldehyde for 10 min. Five pictures per sample were acquired randomly using an Axioimager D1 upright fluorescence microscope (Leica Biosystems, Nußloch, Baden-Wurttemberg, Germany), and the relative amount of fluorescence was determined by FIJI (ImageJ) https://imagej.net/Fiji, accessed on 10 March 2022.

### 2.19. Intracellular ATP Determination

Intracellular ATP levels were determined in primary hepatocytes by using an ATPlite^TM^ luminescence ATP detection kit (Perkin Elmer, Waltham, MA, USA) following manufacturer’s recommendation. In brief, 50 μL of the mammalian cell lysis solution were added per well with 100 μL of MEM without serum and incubated on an orbital shaker (700 rpm, 5′, RT). Then, 35 μL of the lysate were incubated in the white plate’s wells (Gibco, ThermoFisher Scientific) containing 100 μL of MEM 0% previously added, and 50 μL of the substrate solution were then added and incubated (700 rpm, 5′, RT). The plate was adapted to the dark for 10′ and the luminescence was measured in a luminometer. The obtained values were normalized to total protein concentration.

### 2.20. Carbamoyl-Phosphate 1 Synthetase (CPS1) Activity Determination

The amount of hepatic CPS1 activity was determined by using a previously described method [25]. Briefly, 20 mg liver samples were lysed in 200 μL of mitochondrial lysis buffer (10 mM HEPES pH 7.4, 0.5% triton X-100 and 2 mM DTT) and samples normalized to 0.1 μg/μL. Next, 10 μL of sample were mixed with 90 μL and then incubated with 25 μL 50 mM ornithine, 25 μL 2.7M triethanolamine and 25 μL 150 mM carbamoylphosphate. A 0 to 100 nmoles standard curve was also incubated with the ornithine: triethanolamine carbamoylphosphate mixture. After 30 min at 37 °C incubation protected from light, samples were added 80 μL phosphoric acid: sulfuric acid 3:1 and 20 μL butanedione monoxime. The plate was shacked for 30 s and then incubated at 95 °C during 30 min prior absorbance determination at 490 nm wavelength.

### 2.21. Ornithine Transcarbamylase (OTC) Activity Determination

To start, 20 mg liver samples were lysed in 200 μL of mitochondrial lysis buffer (10 mM HEPES pH 7.4, 0.5% triton X-100 and 2 mM DTT) and samples normalized to 2 μg/μL. A 20 μL sample was incubated at 37 °C during 10 min with 40 μL of an enzymatic reaction mixture containing 10 μM NH_4_HCO_3_, 1 μM ATP, 2 μM magnesium acetate, 1 μM N-acetyl-L-glutamic, 0.2 μM DTT and 10 μM triethanolamine. A standard curve of 60 μL from 0 to 50 nmol was also incubated. Then, 6 μL of 2M hydroxylamine were added and the plate was incubated at 95 °C for 10 min. Next, 240 μL were added of a stop/developer solution consisting on equal parts of 850 mg antipyrine in 100 mL of 40% sulfuric acid and 625 mg of 2,3-butanedione monoxime in 5% acetic acid. The plate was incubated at 95 °C for 15 min and absorbance was determined at 450 nm wavelength.

### 2.22. Metabolic Flux Determination in Primary Hepatocytes

For the relative quantification with high resolution chromatography-coupled Time-of-Flight mass spectrometry analysis in primary hepatocytes, 30 μg/mL of L-methionine (^13^C5, 99%, Cambridge Isotope Laboratories, Inc.) were added to methionine-free cell media (ThermoFisher Scientific); when the experiment was finished, plates were immediately frozen and the subsequent analysis was performed. For the metabolite extraction, 500 μL ice cold methanol (80%) containing 200 mM acetic acid was added to the wells of the culture plates, then, buffer was transferred to Eppendorf tubes and centrifuged at 3750 rpm at 4 °C during 30 min. After, 200 μL of the chilled supernatants were evaporated with a SpeedVac in approximately 1.5 h. The resulting pellets were resuspended in 150 μL water/acetonitrile (MeCN) (40/60/*v*/*v*). Measurements were made with a UPLC system (Acquity, Waters Inc.) coupled to a Time-of-Flight mass spectrometer (ToF MS, SYNAPT G2, Waters Inc.). A 2.1 × 100 mm, 1.7 μm BEH amide column (Waters Inc.), thermostated at 40 °C, was used to separate the analytes before entering the MS. Solvent A (aqueous phase) consisted of 99.5% water, 0.5% formic acid and 20 mM ammonium formate while solvent B (organic phase) consisted of 29.5% water, 70% acetonitrile, 0.5% formic acid and 1mM ammonium formate. Samples were injected following a gradient, and every eight injections, a QC sample was injected.

### 2.23. Subcellular Protein Extraction

Cytosolic, membrane and nuclear fractions’ lysates from frozen liver tissue samples were obtained using the Subcellular Proteome Extraction Kit (Calbiochem) following manufacturer’s procedure. The lysates were quantified by BCA protein assay (ThermoFisher Scientific). Mitochondrial isolation from frozen liver tissue samples was performed using the Mitochondria/Cytosol Fractionation Kit (Abcam, Cambridge, UK) as indicated by the manufacturer. Briefly, frozen livers were ground in mortar previously cooled with liquid nitrogen. Then, they were resuspended in the cytosolic buffer and mechanically homogenized, always in cold. Cytosols were centrifuged (13,000 rpm, 10′) three times. Pellets obtained from the sequential centrifugations were collected as crude mitochondria and finally mixed and lysed in the mitochondrial buffer for BCA quantification.

### 2.24. Statistical Analysis

All the experiments were performed at least in triplicate, with *n* = 3 (in vitro) and *n* = 4 (in vivo). The data are expressed as mean ± SEM and represent the fold change vs. control mean value when indicated. Statistical significance was determined using Prism 8 (GraphPad Software, Dotmatics, Bishop’s Stortford, East Hertfordshire, UK). Groups were compared by one-way analysis of variance (ANOVA) followed by post hoc Bonferroni tests (for three or more groups) or Student´s t-tests (for two groups). Correlations were calculated by using Pearson’s correlation coefficient from Prism 8 (GraphPad Software).

## 3. Results

### 3.1. Altered Nitrogen Homeostasis, Polyamine Synthesis and Methionine Cycle during Drug-Induced Liver Injury

Considering that hyperammonemia is a common feature in liver diseases, the homeostasis of the cation was evaluated in mice administered 360 mg/kg APAP for 48 h. This animal model is widely used for the study of DILI, as it mimics the phenotype observed in humans [26,27] with the characteristic appearance in necrotic areas, as observed in Figure 1A. In concordance with previous studies, an increased ammonia content was also observed in mice administered APAP (Figure 1B), while GLUL expression as an ammonia-scavenging pathway was depleted at both protein (Figure 1C) and mRNA levels (Figure 1D). As previously mentioned, ammonia scavenging in hepatocytes is mediated by GLUL and the mitochondrial urea cycle, with carbamoylphosphate synthase 1 (CPS1) and ornithine transcarbamylase (OTC) as limiting enzymes [28]. Surprisingly, mRNA expression of arginase *1* (*Arg1*), *Cps1* and *Otc* was found to be increased in APAP-administered mice (Figure 1D), together with an increased ammonia production by the high-affinity isoform 1 of *glutaminase* (*Gls1*) [29].

As mentioned previously, there is an interconnection between ammonia homeostasis and other pathways, such as the methionine cycle [22] or polyamine synthesis [23], in which decarboxy-S-adenosyl-methionine (dc-AdoMet) and ornithine are converted into spermidine (Spd) and spermine (Spm) (Figure 1E). The amount of these metabolites present in both pathways was evaluated, observing a decrease in the polyamines Spd and Spm, despite increased dc-AdoMet levels (Figure 1F). A lower amount of AdoMet was also observed in mice treated with APAP (Figure 1F), suggesting alterations not only in nitrogen metabolism but also in the methionine cycle.

Such alterations in the methionine cycle have been previously reported during liver diseases [19,20]. The lower AdoMet content observed in vivo during DILI prompted us to measure the in vitro GNMT expression as a key limiting enzyme of the methionine cycle. Isolated murine wild-type hepatocytes were cultured for different periods of APAP stimulation prior to collection, observing a time-dependent decrease in GNMT protein expression (Figure 2A). Correlated with this, the in vivo characterization also showed a GNMT downregulation at both the protein (Figure 2B) and mRNA levels (Figure 2C), with a tendency towards decreased methionine synthase (*Mtr*) mRNA expression. Interestingly, the MATI/III protein upregulation observed in DILI may attempt to compensate for the reduction in AdoMet levels in DILI (Figure 2B), while the increased mRNA expression of the folate’s cycle enzymes *Methylenetetrahydrofolate reductase* (*Mthfr*) and *synthase* (*Mthfs*) suggested an enhanced activity of this pathway (Figure 2C).

Overall, these findings point out that the alterations in ammonia homeostasis during DILI are accompanied by a perturbed polyamine synthesis pathway and alterations in the methionine cycle, with decreased AdoMet levels and GNMT expression as hallmarks.

### 3.2. miR-873-5p Overexpression Leads to Perturbations in Methionine Cycle during Drug-Induced Liver Injury

The role of miR-873-5p in inhibiting GNMT expression, and thus decreasing methionine cycle activity, has been characterized in detail in our laboratory [17,18]. Taking this into account, we hypothesized that miR-873-5p might also play a role in DILI by leading to observed GNMT downregulation (Figure 2A–C). Consistent with these findings, miR-873-5p levels in primary wild-type hepatocytes under APAP stimulation (Figure 2D) and in liver from APAP-administered C57BL/6J mice (Figure 2E) showed an upregulation. Likewise, circulating miR-873-5p was also observed to be increased in samples from DILI patients (*n* = 10) compared to healthy individuals (*n* = 10) (Figure 2F, Table 1). Remarkably, a positive correlation was also found between the circulating levels of the micro-RNA and other liver disease markers such as aspartate aminotransferase (AST) and alanine aminotransferase (ALT). The correlation between miR-873-5p and alkaline phosphatase (ALP) reflected a tendency, whereas it did not correlate with total bilirubin levels (TBL) (Figure 2G).

miR-873-5p contribution to DILI-induced GNMT downregulation was further addressed in vitro in primary wild-type hepatocytes by an anti-miR therapy, which effectively decreased the expression of the miRNA (Figure 3A). Subsequently, the loss of *Gnmt* mRNA (Figure 3B) and protein expression (Appendix A) was partially prevented by the anti-miR-873-5p, thus leading to decreased APAP-induced cell death, measured by TUNEL at different times of 3, 6 and 9 h (Figure 3C) and Trypan Blue at 6 h (Appendix A). Remarkably, anti-miR-873-5p prevented the GNMT loss at both cytoplasmic and nuclear levels (Appendix A). Otherwise, the miR-873-5p enhanced expression by a mimic-miRNA further aggravated hepatotoxicity, as shown in Appendix A, while GNMT overexpression by a transient transfection of wild-type primary hepatocytes also led to protection against APAP at 3 and 6 h but not as much as at 9 h (Figure 3D and Appendix A). The relevance of GNMT expression in APAP hepatotoxicity and the protective effect exerted by the anti-miR was further determined in primary hepatocytes lacking *Gnmt*, where, although miR-873-5p was inhibited (Appendix A), no protection was observed (Appendix A).

An in vivo preclinical approach by administering the anti-miR-873-5p to C57BL/6J mice demonstrated the relevance of the miRNA and its modulation of GNMT expression. First, it was determined that the treatment with a miR-Ctrl did not modulate the DILI phenotype in terms of necrotic areas (Appendix A), ammonia accumulation (Appendix A) or GLUL expression (Appendix A). Then, mice were administered 360 mg/kg, and 24 h afterwards, a group was treated with 60 μg anti-miR-873-5p per mouse (Appendix A). As a result of miR-873-5p expression inhibition (Figure 3E), GNMT restored its expression at the protein (Figure 3F) and mRNA levels (Appendix A), suggesting a recovery of the methionine cycle. Contrary to the effect observed under APAP, where mRNA from folate’s cycle enzymes was overexpressed (Figure 2C), it was downregulated by the anti-miR (Appendix A). Remarkably, GNMT restoration under miR-873-5p knockdown led to the amelioration of the DILI phenotype, with a decreased appearance of necrotic areas in treated mice (Figure 3G) and a reduced content of ALT and AST as indicators of liver damage (Figure 3H).

Altogether, the presented results highlight the relevance of miR-873-5p in modulating GNMT expression and the methionine cycle in DILI. The amelioration of the phenotype by the anti-miR-based therapy further demonstrates these findings.

### 3.3. Targeting miR-873-5p Ameliorates the Inflammatory Response and Mitochondrial Dysfunction in DILI

Under APAP overdose, the production of N-acetyl-p-benzoquinone imine (NAPQI) promotes glutathione depletion that leads to mitochondrial dysfunction and subsequent necrosis, triggering an enhanced inflammatory response in hepatocytes [30,31]. Once having observed the effectivity of the anti-miR-873-5p in preventing DILI, the inflammatory response was analyzed. Macrophage infiltration determined by F4/80 staining (Figure 4A) and the presence of tumor necrosis factor (TNF) in serum (Figure 4B), widely associated with liver injury [32], were evaluated, observing a decrease in both markers of inflammation. Correlated with this, hepatic superoxide dismutase (SOD) activity was also decreased (Figure 4C), together with the mRNA expression of inflammatory markers (Figure 4D) or phosphorylated c-Jun N-terminal kinase 1/2 (JNK 1/2) protein levels in the mitochondria (Appendix A) as another indicator of oxidative stress. Remarkably, the amount of reduced glutathione compared to oxidized glutathione (GSH/GSSG) under APAP overdose was increased in those mice treated with the anti-miR (Figure 4E), in spite of the decreased mRNA expression of enzymes from the glutathione synthesis pathway (Appendix A). Otherwise, the mRNA expression of *cystathionine β-synthase* (*Cbs*) from the transsulfuration pathway was enhanced (Appendix A).

The preventive effect of inhibiting miR-873-5p expression on the inflammatory response was also evaluated in vitro in primary hepatocytes treated with APAP at different times of 3 and 6 h. Correlated with the observed results in vivo, the amount of mitochondrial reactive oxygen species production (ROS) was decreased, as measured by mitoSOX (Figure 4F), while APAP-induced ATP depletion with time was also prevented (Figure 4G) in the anti-miR-873-5p group. Moreover, the relative content of mitochondria, also reduced by APAP time-dependently, was also increased in the treated group (Figure 4H and Appendix A). An increased oxygen consumption rate was observed in the anti-miR group, measured by Seahorse XF-24 analyzer at different stages: (i) basal respiratory activity, (ii) ATP-linked after treating cells with oligomycin and (iii) maximal respiration after carbonyl cyanide 4-(trifluoromethoxy)phenylhydrazone (FCCP) treatment (Figure 4I). The ATP production rate by either glycolysis or oxidative phosphorylation was also determined. Interestingly, as shown in Figure 4J, those hepatocytes blocking miR-873-5p expression showed an increase in ATP production by mitochondria respiration with a reduction in glycolysis in comparison to those hepatocytes only stimulated with APAP with a reduction in ATP production.

The present results suggest that inhibition of miR-873-5p reduces the inflammatory response under APAP both in vivo and in vitro, restores GSH levels and improves mitochondrial functionality, thereby reducing the overproduction of ROS.

### 3.4. miR-873-5p Inhibition Rewires Ammonia Homeostasis and Restores Polyamine Synthesis

Considering the obtained results that suggest an improved mitochondrial functionality and respiration when miR-873-5p expression was inhibited during APAP overdose, the hepatic ammonia content was determined in mice treated with the anti-miR therapy. As shown in Figure 5A, the ammonia content in the liver from treated mice was reduced, together with the content of the cation in the serum (Appendix A) as a possible trigger of other complications such as hepatic encephalopathy (HE) [33]. When evaluating ammonium scavenging pathways, an increased GLUL expression at protein levels (Figure 5B) was observed, whereas mRNA *Glul* expression tended to increase (Figure 5C), suggesting that GLUL restoration might only occur in the perivenous region. Although anti-miR-873-5p-mediated GLUL restored expression might place its gene as a downstream target of the miR-873-5p, *Glul* mRNA expression is not increased in wild-type hepatocytes treated with the anti-miR, but is overexpressed in those hepatocytes lacking *Gnmt* (Appendix A). *Glul* expression regulation appears to be modulated by a mechanism independent from miR-873-5p.

Surprisingly, the mRNA expression (Figure 5C) and activity from CPS1 and OTC (Figure 5D) from the urea cycle were reduced in the anti-miR treated group. Thus, detoxification of ammonia content might be due to GLUL recovery or reduced hepatocyte death without an enhancement of the urea cycle. Likewise, a reduced mRNA expression of both *Gls1* and *Gls2* isoforms, responsible for producing ammonia, was also observed in anti-miR-treated mice (Figure 5C). The interconnection between the urea cycle and other metabolic pathways prompted us to evaluate the hepatic content of certain metabolites from the methionine cycle (AdoMet and dc-AdoMet) and polyamines (Spm and Spd), which were altered under APAP overdose (Figure 1F). As observed in Figure 5E, the possible methionine cycle restoration led to a AdoMet increase caused by the anti-miR-873-5p. Although hepatic dc-AdoMet did not show any variation, hepatic Spd levels in treated mice suggested the promotion of polyamine synthesis (Figure 5E), despite the decreased mRNA expression of enzymes involved in the pathway (Appendix A).

The possible rewiring in metabolic flux was characterized in a fluxomic experiment where primary wild-type hepatocytes were treated with APAP for 1 and 3 h and stimulated with labelled ^13^C-methionine for measuring metabolites using mass spectrometry. As observed in Figure 6A, hepatocytes under anti-miR-873-5p therapy displayed increased labelled methionine levels accompanied by an increased AdoMet content. The increased MTA^+1^ may imply a recovered methionine re-synthesis, whereas the decrease in labelled dc-AdoMet and increased Spd and Spm might further demonstrate increased polyamine synthesis. In summary, these findings connect the possible recovery of the methionine cycle with a higher ornithine availability due to a decreased urea cycle, which would promote polyamine synthesis.

In order to further demonstrate the hypothesis of a decreased urea cycle and a subsequent increased polyamine synthesis, both pathways were modulated (Figure 6B). First, hepatocytes were stimulated with ammonia in order to scavenge the urea cycle with a non-toxic concentration of ammonium chloride [34]. Interestingly, the protective effect of the anti-miR-873-5p was lost dose-dependently in those hepatocytes treated with ammonium chloride (Figure 6C and Appendix A). Similarly, the inhibition of AdoMet decarboxylation by SAM486A and ornithine decarboxylation by DFMO [35] also led to the absence of the protective effect exerted by the anti-miR (Figure 6D and Appendix A).

Taken together, these results present the interconnection between ammonia homeostasis and the methionine cycle and enhanced polyamine synthesis, responsible for preventing hepatotoxicity by inhibiting miR-873-5p expression.

### 3.5. The Proliferative Response is Increased When Targeting miR-873-5p

Finally, the hepatic proliferative response was evaluated, considering: (i) the increased ATP levels in primary hepatocytes when targeting miR-873-5p (Figure 4I), essential for liver regeneration after an injury [36,37] and (ii) the role of polyamines in cell regeneration [25]. The in vitro mRNA determination of the proliferation markers *Proliferating cell nuclear antigen* (*Pcna*) and *Cyclin D1* showed an increased expression under miR-873-5p inhibition (Figure 7A). Likewise, the hepatic PCNA and cyclin D1 expression at protein levels showed an increase under APAP stimulation, attempting to compensate for the hepatotoxic damage, which is further enhanced by the anti-miR-873-5p treatment (Figure 7B). The nuclear amount of these markers, together with β-catenin, also increased in the liver from anti-miR-treated mice (Figure 7C). Finally, the hepatic mRNA expression of proliferative markers such as *β-catenin, Pcna* or *Cyclin B, D1* and *E* also suggested an increased proliferative response in treated mice (Figure 7D).

In summary, the prevention of APAP hepatotoxicity by the anti-miR-873-5p may also be due to an enhanced proliferative response derived from increased polyamine synthesis and bioenergetics restoration.

## 4. Discussion

This work highlights the relevance of the interconnection between nitrogen metabolism, with the urea cycle and polyamine synthesis pathways, and the methionine cycle in DILI development. Related to the maintenance of ammonia in the liver, hepatocytes play a key role in scavenging the cation by either: (i) the action of CPS1 from the urea cycle in the periportal region or (ii) the GLUL-mediated glutamine synthesis in the perivenous region [1]. Although the liver achieves a normal homeostasis of the cation, during liver pathologies, perturbations in both ammonia production and/or scavenging lead to hyperammonemia [38,39]. Indeed, in a previously developed staining method for ammonia, our group already characterized hepatic hyperammonemia in chronic and acute liver diseases [4]. Correlated with this, the present study characterizes the accumulation of the cation in those mice treated with 360 mg/kg APAP for 48 h accompanied by a GLUL depletion, already reported to occur [40]. Otherwise, the increased mRNA expression of the urea cycle-limiting enzymes *Cps1* and *Otc* in those mice administered APAP suggested enhanced urea cycle activity that might attempt to compensate the hepatotoxic ammonia accumulation during DILI.

As cited above, there is an existing connection between the urea cycle and other pathways such as polyamine synthesis and the methionine cycle [22,23]. Related to this, the metabolomics determination of polyamines Spd and Spm revealed a decrease in those mice under APAP, together with a decreased hepatic AdoMet content as characterized in previous studies [41]. AdoMet has already been reported as an indicator of the methionine cycle activity [41], which appeared to be downregulated during DILI. AdoMet anabolism is mediated in the liver by MATI/III, while several studies have already provided evidence for the formation of peroxynitrite during APAP hepatotoxicity [42] with S-nitrosylation of MATI/III associated with marked inactivation of the enzyme. Additionally, MATI/III inactivation is reversed by increased levels of GSH [43]. Although our data revealed overexpression of MATI/III in mice overdosed with APAP, lower AdoMet levels indicated an inhibition of this enzyme, which potentially could be reversed when oxidative stress and GSH levels are restored under anti-miR-873-5p treatment. GNMT, responsible of AdoMet catabolism, was downregulated in our study under APAP overdose according to previous findings already reported in many chronic liver pathologies. Indeed, mice lacking the gene spontaneously develop steatosis and hepatocellular carcinoma [19]. Thus, the increased hepatic dc-AdoMet content in DILI observed in our study may be a consequence of the combination of a reduced AdoMet metabolism and decreased polyamine synthesis, where Spm has been reported to negatively regulate AdoMet decarboxylase [44]. These results point towards not only perturbations in ammonium homeostasis during DILI, but also a decreased methionine cycle activity. The recovery of the methionine cycle may be essential to diminish the hepatotoxic effects of APAP during DILI development.

This study focuses on the miR-873-5p-mediated regulation of GNMT expression for restoring the methionine cycle and preventing the pathology. This miRNA has already been reported by our group to act as an epigenetic modulator of GNMT, finding it overexpressed in NAFLD [17] and fibrosis [18] and being inversely correlated with the expression of the enzyme. Similar to previous findings, GNMT downregulation was accompanied by an increased miR-873-5p expression in both primary wild-type hepatocytes under different periods of APAP overdose and liver from C57BL/6J mice treated with APAP. Remarkably, the presence of the miRNA was found in the serum from DILI patients, being increased when compared to healthy individuals and finding it correlated with ALT and AST and liver damage parameters.

The GNMT restoration, according to protein levels and mRNA expression, provided by an anti-miR-873-5p-based therapy prevented the development of DILI in primary wild-type hepatocytes at different times of APAP, while in treated mice it reduced the appearance of necrotic areas in the liver and transaminases levels in the serum. On the contrary, the overexpression of miR-873-5p by using a mimic-miRNA aggravated the hepatotoxic effect of APAP overdose. Although the loss of *Gnmt* mRNA was not completely prevented by the anti-miRNA, it must be taken into account that the epigenetic regulation of *Gnmt* is not only determined by the miR-873-5p but also by DNA hypermethylation. This plays an important role in the repression of GNMT, at least in liver cancer, meaning that the changes by the anti-miR-873-5p were expected to be mild [45]. Nevertheless, the protein determination in vitro by Western blot and immunofluorescence revealed a significant increase in the anti-miR-873-5p-treated groups at different times and at both cytoplasmic and nuclear levels. The importance of GNMT restoration by anti-miRNA treatment was further demonstrated in primary hepatocytes isolated from mice lacking *Gnmt* (*Gnmt^-/-^*), where the protective effect of anti-miR therapy was lost. Likewise, the transient GNMT overexpression in primary wild-type hepatocytes also protected the cells from APAP at 3 and 6 h of incubation. However, this effect over a longer period (9 h) was not achieved, highlighting the efficacy of miRNA-based therapies as an improved therapeutic approach for DILI [15,46].

Apart from the appearance of necrotic areas in the liver, another DILI hallmark is the development of mitochondrial dysfunction and a pro-inflammatory stage due to glutathione depletion by produced NAPQI [30,31]. The anti-miR873-5p therapy was also effective in reducing macrophage infiltration, measured by F4/80 staining and inflammation development. The presence of TNF in the serum and SOD activity were also reduced by the therapy, together with a reduced mRNA expression of pro-inflammatory cytokines. Although TNF might be essential for liver regeneration [47], in the context when it is measured, when liver regeneration has already occurred, such reduction in the serum might be an indicator readout of the reduced injury under the therapy [32]. The characteristic glutathione depletion was also prevented by the anti-miR-873-5p, as reflected by the observed increase in the GSH/GSSG ratio. The mRNA expression of glutathione synthesis enzymes suggested that GSH restoration was not due to an enhanced production, but to reduced oxidative stress development. This was further confirmed by mitoSOX staining, where ROS production increased upon APAP stimulation and was reduced by the anti-miR-873-5p at different times. Mitochondrial functionality was improved by the therapy, as reflected in the increased MitoTracker staining that was accompanied by the prevention of ATP loss by APAP. The OCR measurement also showed an increased respiration of primary hepatocytes treated with the anti-miRNA, where the ATP source was mainly from oxidative phosphorylation. The interaction with GNMT, restored under anti-miR-873-5p and mitochondrial Complex II [17], might lead to the observed reduction in mitochondrial dysfunction independently of other key antioxidant enzymes, as is shown in the case of SOD activity [48].

Considering the increased ammonium content and the loss of GLUL expression in the perivenous region upon APAP overdose, the restoration of ammonium levels in both the liver and serum and GLUL protein expression may be a consequence of the observed reduction in mitochondrial dysfunction and subsequent hepatocyte protection. Indeed, hyperammonemia has already been linked to mitochondrial dysfunction and, in the meantime, reported to induce, among others, a senescence phenotype [49]. The observed GLUL recovery may suggest miR-873-5p as an upstream regulator of its gene. However, the mRNA determination in primary hepatocytes did not show an increased expression under anti-miR-873-5p therapy in wild-type cells, whereas *Gnmt^-/-^* did. Thus, the *Glul* expression cannot be proposed to be directly regulated by miR-873-5p, as its expression might be indirectly modulated by the damage suffered by the hepatocyte. Likewise, the reduced ammonium content could also be a consequence of reduced cell death, as urea cycle enzymes’ *Cps1* and *Otc* mRNA expression and activities were reduced by the anti-miR-873-5p, despite reducing hepatic hyperammonemia. Considering this fact, the less active urea cycle might lead to an increased ornithine availability, which would be then converted into putrescine and act as a substrate, together with dc-AdoMet, for the synthesis of polyamines. This could explain the increased Spd levels in the liver from mice treated with the anti-miR-873-5p, a metabolite with a role in improving mitochondrial functionality in other cell types [50,51] and reported to be essential for hepatocyte proliferation [52]. The rewiring in the hepatocyte metabolism from the urea cycle, increased under APAP overdose, towards polyamine synthesis for hepatocyte protection was further confirmed by the fluxomic approach performed. The administration of labelled methionine to hepatocytes resulted in an increased Spd, Spm and Spm-NAc content with decreased dc-AdoMet levels. The recovery of methionine cycle when targeting miR-873-5p is depicted by both: (i) increased GNMT expression at protein and mRNA levels and (ii) hepatic AdoMet restoration observed by metabolomics. Thus, AdoMet restoration may be due to the recovered methionine cycle, while the increased *Mtr* mRNA expression observed by the anti-miR-873-5p suggested an increased homocysteine recycling and subsequent higher methionine levels. The scavenging of the urea cycle by increasing ammonium content in culture medium prevented the hepatoprotective effect by the anti-miR-873-5p therapy, while the inhibition of polyamine synthesis had the same effect, thus demonstrating the relevance of the proposed metabolic shifting with the therapy.

As a consequence of the decreased urea cycle and higher ornithine availability leading to increased polyamine synthesis, the proliferative response of hepatocytes was increased. This was observed in vitro by the higher mRNA expression of the proliferation markers *Pcna* and *Cyclin D1*. Correlated with this, the tissue expression of both markers and mRNA expression was also higher in liver from mice treated with the anti-miR-873-5p. Although, during APAP overdose, there is a higher expression in an attempt to compensate for the hepatotoxic damage, this may not be sufficient to reduce the appearance of necrotic areas. Moreover, in isolated fractions of liver tissue from treated mice, an increased nuclear expression of such proliferation markers and β-catenin was also observed. The mRNA expression of other proliferation markers, such as *Cyclin B* and *E* isoforms, was also higher in the anti-miR-treated group. Therefore, the role that polyamines have in mitochondrial functionality [50,51] and hepatocyte proliferation [52], together with the requirements of adequate ATP levels in liver regeneration [36,37], may enhance the regenerative response for preventing APAP-derived hepatotoxicity.

In summary, the present work demonstrates the preventive effect of an anti-miR-873-5p-based therapy on the hepatotoxic effect of APAP overdose. The methionine cycle recovery due to the increased GNMT expression when targeting the miRNA leads to an enhanced mitochondrial functionality, thus diminishing the effect of APAP. Meanwhile, the reduced ammonium accumulation may lead to reduced urea cycle activity and a subsequent ornithine availability. In combination with the recovery of hepatic AdoMet content, ornithine might be the substrate of the enhanced polyamine synthesis observed when targeting the miRNA. The role that polyamines have in mitochondrial functionality and liver regeneration might modulate the regenerative response upon APAP overdose while preventing the cytotoxic effect of the compound.

## Figures and Tables

**Figure 1 antioxidants-11-00897-f001:**
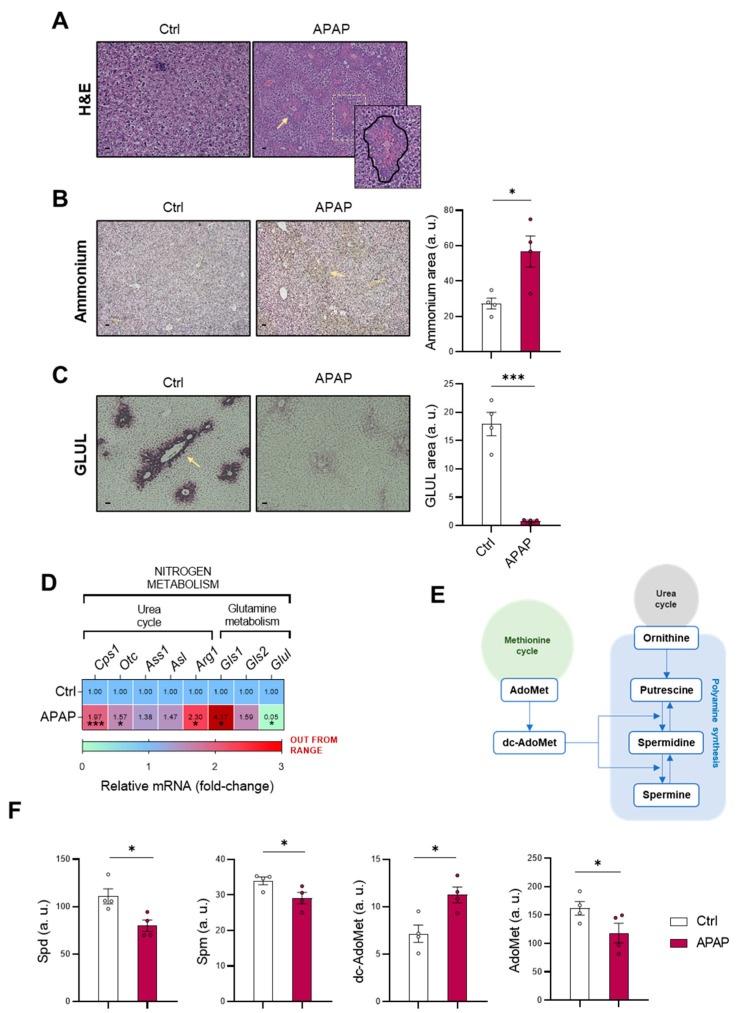
Ammonia homeostasis, methionine cycle and polyamine synthesis are altered in DILI. (**A**) Hepatic hematoxylin and eosin (H&E) staining (zoom-in); representative micrographs and respective quantification of (**B**) ammonia and (**C**) glutamine synthetase (GLUL) staining (highlighted with yellow arrows); (**D**) relative mRNA expression of the urea cycle and glutamine metabolism genes and (**F**) determination of spermidine (Spd), spermine (Spm), decarboxy-S-adenosylmethionine (dc-AdoMet) and S-adenosylmethionine (AdoMet) in liver from mice treated with 360 mg/kg acetaminophen (APAP) compared to healthy controls. (**E**) Schematic representation of interconnection among the methionine and urea cycles and polyamine synthesis. Scale bar corresponds to 50 μm. * *p* < 0.05 and *** *p* < 0.001 are shown.

**Figure 2 antioxidants-11-00897-f002:**
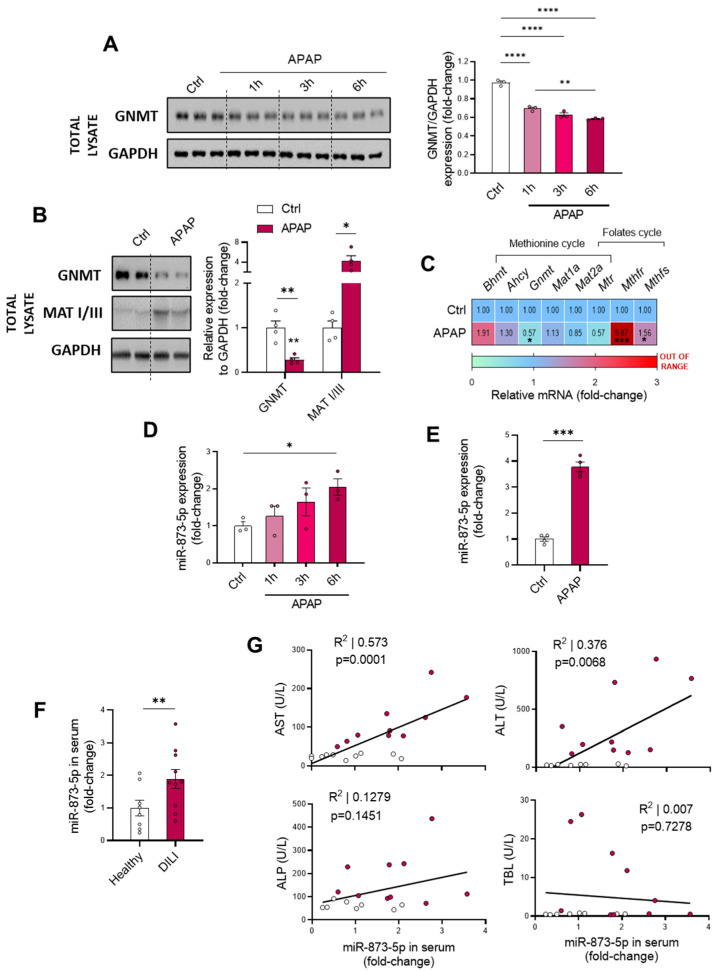
Glycine-N-methyltransferase (GNMT) expression is decreased in DILI by the enhanced expression of miR-873-5p. (**A**) Western blot of GNMT, loading control and respective quantification in primary hepatocytes cultured for 1, 3 and 6 h with 10 mM acetaminophen (APAP); (**B**) Western blot of GNMT and Methionine adenosyltransferase I/III (MATI/III) and relative expression of (**C**) mRNA from methionine and folates cycles enzymes in liver from mice administered 360 mg/kg APAP for 48 h. miR-873-5p expression in (**D**) primary hepatocytes cultured for 1, 3 and 6 h with APAP, (**E**) Liver from mice administered 360 mg/kg APAP for 48 h and (**F**) serum from drug-induced liver injury patients (DILI) compared to a control group or healthy individuals. (**G**) Correlation between levels in serum of miR-873-5p and aspartate aminotransferase (AST), alanine. aminotransferase (ALT), alkaline phosphatase (ALP) and total bilirubin (TBL) from DILI patients compared to healthy individuals. * *p* < 0.05, ** *p* < 0.01, *** *p* < 0.001 and **** *p <* 0.0001 are shown.

**Figure 3 antioxidants-11-00897-f003:**
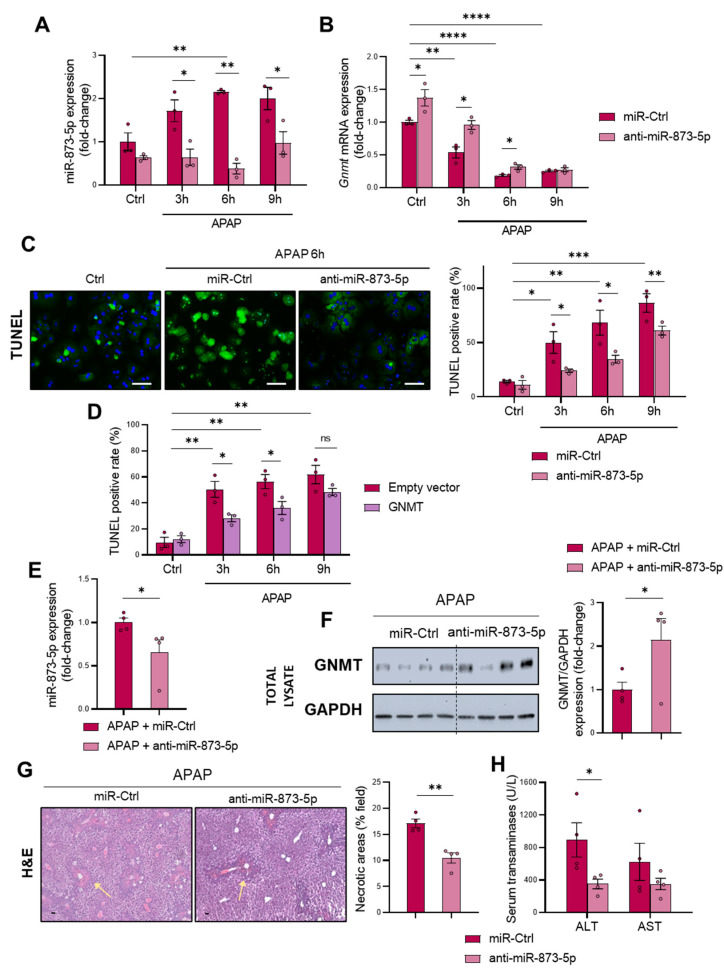
Targeting miR-873-5p restores glycine-N-methyltransferase (GNMT) expression protecting hepatocytes from DILI (**A**) miR-873-5p and (**B**) *Gnmt* expression and (**C**) representative micrographs and respective quantification of TUNEL staining in primary hepatocytes after 3, 6 and 9 h of 10 mM acetaminophen (APAP) stimulation and treated with an anti-miR against miR-873-5p (anti-miR-873-5p) or an unrelated control (miR-Ctrl); (**D**) TUNEL determination in primary hepatocytes stimulated with 3, 6 and 9 h APAP and treated with a GNMT expression or an empty vector; (**E**) Hepatic miR-873-5p and (**F**) Western blot determination of GNMT and respective determination related to a loading control, (**G**) representative micrographs of hematoxylin and eosin (H&E) staining and respective quantification of necrotic areas (highlighted with yellow arrows), and (**H**) serum transaminases levels (alanine aminotransferase (ALT) and aspartate aminotransferase (AST) from mice administered 360 mg/kg APAP for 48 h and treated with either anti-miR-873-5p or miR-Ctrl. Scale bar corresponds to 50 μm. * *p* < 0.05, ** *p* < 0.01; *** *p* < 0.001 and **** *p* < 0.0001 are shown. (See also Appendix A).

**Figure 4 antioxidants-11-00897-f004:**
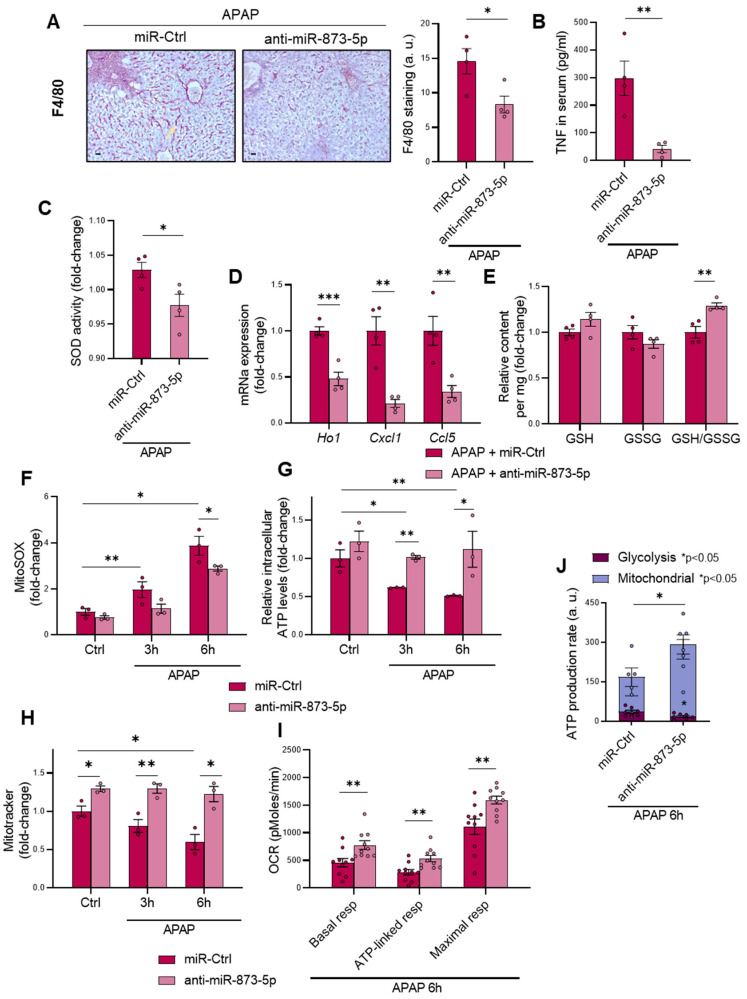
APAP-induced inflammatory response and mitochondrial dysfunction are reduced by the anti-miR-873-5p therapy (**A**) Representative micrographs and respective determination from hepatic F4/80 staining (highlighted with yellow arrows), (**B**) serum levels of tumor necrosis factor (TNF)**,** (**C**) relative hepatic superoxide dismutase (SOD) activity, (**D**) hepatic mRNA expression of proinflammatory cytokines and (**E**) relative hepatic reduced glutathione (GSH), oxidized glutathione (GSSG) and GSH/GSSG ratio in mice treated with 360 mg/kg acetaminophen (APAP) for 48 h and either anti-miR-873-5p or miR-Ctrl. Scale bar corresponds to 50 μm. (**F**) Relative mitochondrial ROS staining determination by mitoSOX; (**G**). intracellular ATP levels and (**H**) MitoTracker^TM^ staining determination in primary hepatocytes treated with APAP at 3 and 6 h and either anti-miR-873-5p or miR-Ctrl. (**I**) Oxygen consumption rate (OCR) and (**J**) ATP production from glycolysis and mitochondria in primary hepatocytes upon 6 h APAP stimulation and treated with miR-Ctrl or anti-miR-873-5p. * *p* < 0.05, ** *p* < 0.01; *** *p* < 0.001 are shown. (See also Appendix A.).

**Figure 5 antioxidants-11-00897-f005:**
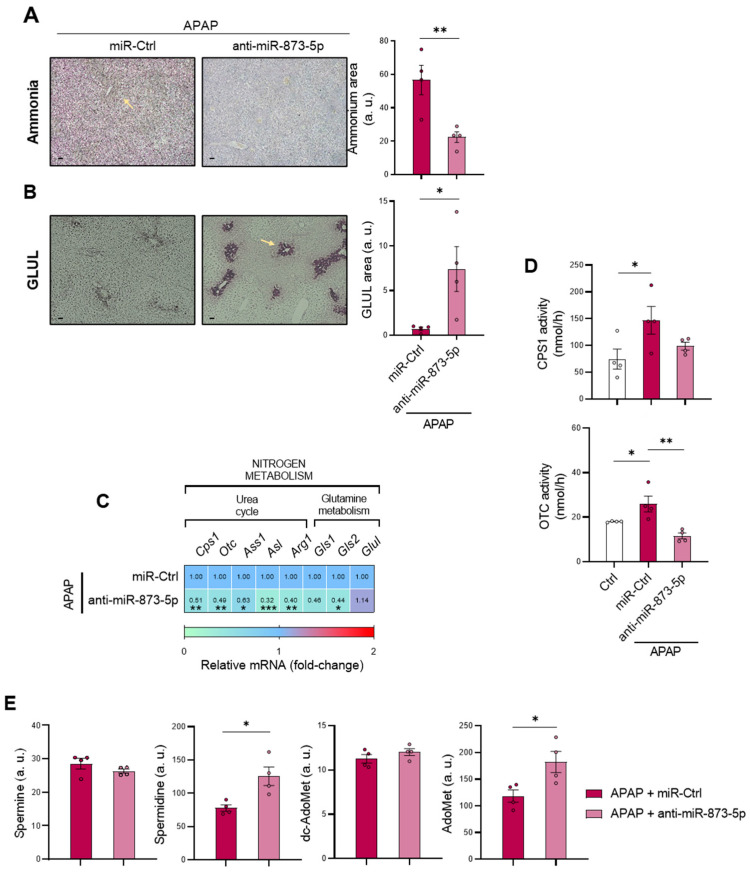
Nitrogen homeostasis, S-adenosylmethionine and polyamines are restored by targeting miR-873-5p. Representative micrographs and respective quantification of (**A**) hepatic ammonia and (**B**) glutamine synthetase (GLUL) staining (highlighted with yellow arrows); (**C**) relative hepatic mRNA expression of urea cycle and glutamine metabolism genes; (**D**) relative hepatic activity of carbamoylphosphate-synthase 1 (CPS1) and ornithine transcarbamilase (OTC) and (**E**) determination of hepatic S-adenosylmethionine (AdoMet), decarboxy-S-adenosylmethionine (dc-AdoMet), spermine (Spm) and spermidine (Spd) in mice administered acetaminophen (APAP) for 48 h and treated with either an anti-miR-873-5p or miR-Ctrl. Scale bar corresponds to 50 μm. * *p* < 0.05, ** *p* < 0.01, *** *p* < 0.001 are shown. (See also Appendix A).

**Figure 6 antioxidants-11-00897-f006:**
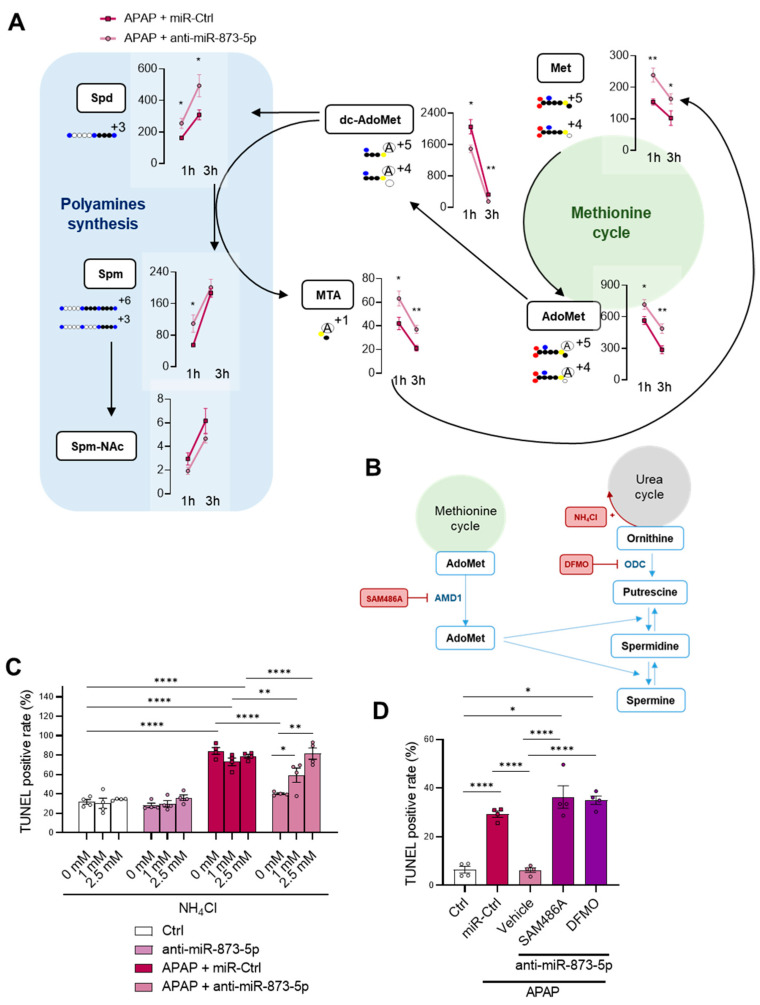
Methionine cycle recovery rewires nitrogen homeostasis from urea cycle to polyamine synthesis. (**A**) Schematic representation and determination of metabolites from methionine metabolism and polyamine synthesis in primary hepatocytes treated with ^13^C–methionine and 10 mM acetaminophen (APAP) at different times (1 and 3 h) and an anti-miR-873-5p or an unrelated control (miR-Ctrl): methionine (Met), S-adenosylmethionine (AdoMet), decarboxy-S-adenosylmethionine (dc-AdoMet), MTA, spermidine (Spd), spermine (Spm) and N-acetylspermine (Spm-NAc). (**B**) Schematic representation of TUNEL determination of primary hepatocytes treated for 6 h with 10mM APAP, an anti-miR-873-5p or miR-Ctrl and (**C**) different concentrations of ammonium chloride (1 mM and 2.5 mM NH_4_Cl) or (**D**) 1 μM SAM486A or 0.5 μM DFMO. * *p* < 0.05, ** *p* < 0.01, **** *p* < 0.0001 are shown.

**Figure 7 antioxidants-11-00897-f007:**
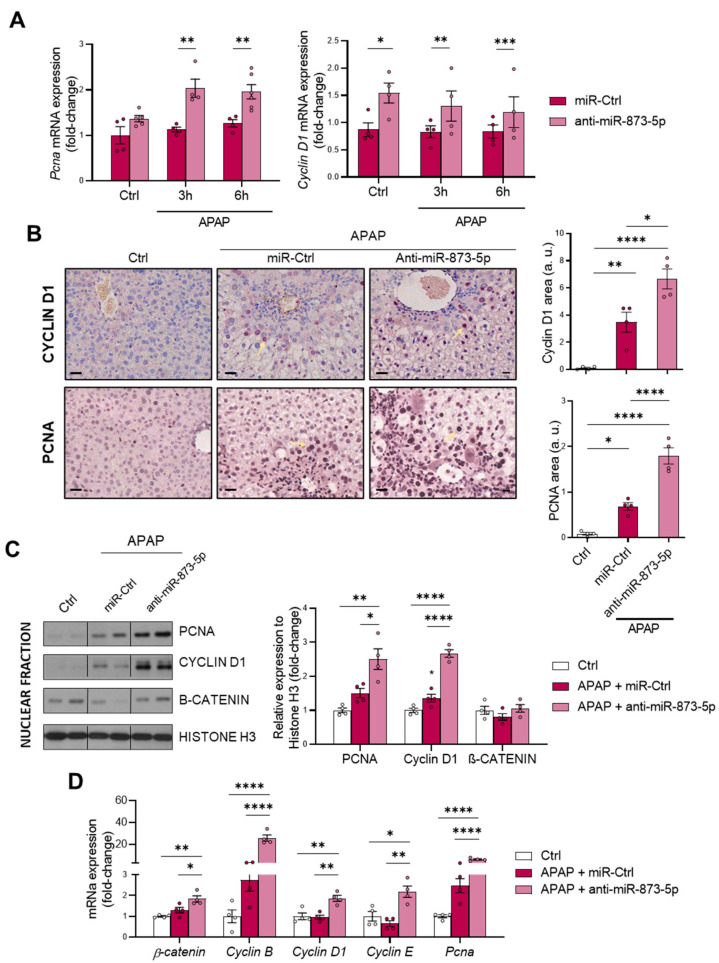
anti-miR-873-5p therapy promotes hepatocyte proliferation for liver regeneration against APAP. (**A**) Relative mRNA expression of *proliferating cell nuclear antigen* (*Pcna)* and *cyclin D1* in primary hepatocytes treated with 10mM acetaminophen (APAP) for 3 and 6 h and either an anti-miR-873-5p or a miR-Ctrl; (**B**) representative micrographs of hepatic cyclin D1 and PCNA stainings (highlighted with yellow arrows) and respective determination; (**C**) Western blot of nuclear fractions of PCNA, cyclin D1 and β-catenin and respective quantification using Histone H3 as a loading control; (**D**) relative hepatic mRNA expression of proliferation genes in mice treated with 360 mg/kg APAP for 48 h and either anti-miR-873-5p or miR-Ctrl. Scale bar corresponds to 50 μm. * *p* < 0.05, ** *p* < 0.01, *** *p* < 0.001, **** *p* < 0.0001 are shown.

**Table 1 antioxidants-11-00897-t001:** Clinical characteristics of drug-induced liver injury (DILI) patients compared to healthy controls.

	N	Age (Years)	Sex (M/F)	AST (U/L)	ALT (U/L)	ALP (U/L)	TBL (U/L)
Healthy	10	47 ± 13	5/5	25 ± 6	23 ± 7	59.1 ± 16	0.7 ± 0.34
DILI	10	44 ± 14	6/4	112 ± 59	375 ± 312	175 ± 113	8.6 ± 10.4

## Data Availability

Data is contained within the article and supplementary material.

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
