# Peer review of "Methionine Cycle Rewiring by Targeting miR-873-5p Modulates Ammonia Metabolism to Protect the Liver from Acetaminophen"

_antioxidants, 2022, doi:10.3390/antiox11050897_

Round 1

Reviewer 1 Report

I read this paper and I think that:

This study is a very interesting, and is considered to be highly needed study for clinical success of the protection of hepatotoxicity during DILI associated to ammonia homeostasis and methionine cycle.

In this article, authors well summarized the GNMT recovery under miR-873-5p knockdown modulates ammonia metabolism, preventing DILI development in hepatocytes. However, there are a few corrections and questions.

1. In line 449, please replace miR-873-5p for miR-873-5p knockdown.

2. In line 469, authors described that “Targeting miR873 5p ameliorates the inflammatory response in DILI”. However, fig.4 showed the not only inflammation but also mitochondrial ROS and ATP production. Please edit title for fig.4 in results. It would be better to replace it with “Targeting miR873 5p ameliorates the inflammatory response and mitochondrial dysfunction in DILI”.

3. In fig.4C, anti-miR-873-5p showed the decrease of SOD activity. Despite decrease of SOD activity, mitochondrial ROS production was decreased. (fig.4F). Please explain the relation of ROS production and SOD activity.

4. In line 43, change to acetaminophen.

5. In materials and methods, describe the approval number for animal and human study.

Author Response

Please find attached in a cover letter the answer to the concerns addressed by both reviewers. We thankfully appreciate the contribution in improving the quality of the manuscript.

Reviewer 2 Report

In this manuscript, authors analyzed the potential biological role of miR-873-5p for the modulation of glycine N-methyltransferase expression and methionine cycle activity under the drug-induced liver injury. Over all, authors did a good job and made a nice logical description. Here only has some minor comments:

  1. Figure 1A right panel, the small graph, authors should clearly show whether or not it zoom-in from which part of the original graph. As if yes, then authors should mention what is it emphasized for in somewhere of text.
  2. All image should have its own scale bar. A lot of photographs missed the scale bar.
  3. In all photograph, it is better using the arrow to point out the focusing observation area/part like as figure 3G.
  4. But there is no any description regarding the “arrows” showed on the figure 3G either in the main text or relative figure legend.
  5. As I know, the symbol of statistic significant different for p value is asterisk (*) but not hashtag (#). I do not agree with using the hashtag to represent the relative statistic p value. I suggest authors should change back the asterisk to represent the relative statistic p value in several figures.

Author Response

(The authors gave the same response as above.)
